# Well-being during COVID-19 pandemic: A comparison of individuals with minoritized sexual and gender identities and cis-heterosexual individuals

**Pichit Buspavanich**[1,2,3☯]*, **Sonia Lech**[1,2,4☯], **Eva Lermer**[5,6], **Mirjam Fischer**[7], **Maximilian Berger**[1,2], **Theresa Vilsmaier**[8], **Till Kaltofen**[8], **Simon Keckstein**[8], **Sven Mahner**[8], **Joachim Behr**[1,9,10], **Christian J. Thaler**[8], **Falk Batz**[8]

**1** Department of Psychiatry, Psychotherapy and Psychosomatics, Brandenburg Medical School Theodor Fontane, Neuruppin, Germany, **2** Department of Psychiatry and Psychotherapy, Charité – Universitätsmedizin Berlin, Corporate Member of Freie Universität Berlin and Humboldt-Universität zu Berlin, Berlin, Germany, **3** Institute of Sexology and Sexual Medicine, Charité –Universitätsmedizin Berlin, Corporate Member of Freie Universität Berlin and Humboldt-Universität zu Berlin, Berlin, Germany, **4** Institute of Medical Sociology and Rehabilitation Science, Charité –Universitätsmedizin Berlin, Corporate Member of Freie Universität Berlin and Humboldt-Universität zu Berlin, Berlin, Germany, **5** Center for Leadership and People Management, LMU Munich, Munich, Germany, **6** FOM University of Applied Sciences of Economics and Management, Essen, Germany, **7** Institute of Sociology and Social Psychology, University of Cologne, Cologne, Germany, **8** Department of Obstetrics and Gynecology and Center for Gynecological Endocrinology and Reproductive Medicine, University Hospital, LMU Munich, Munich, Germany, **9** Research Department of Experimental and Molecular Psychiatry, Department of Psychiatry and Psychotherapy, Charité –Universitätsmedizin Berlin, Corporate Member of Freie Universität Berlin and Humboldt-Universität zu Berlin, Berlin, Germany, **10** Faculty of Health Sciences Brandenburg, Joint Faculty of the University of Potsdam, Brandenburg University of Technology Cottbus-Senftenberg and Brandenburg Medical School, Potsdam, Germany

☯ These authors contributed equally to this work.
* pichit.buspavanich@charite.de

**Data Availability Statement:** All relevant data are within the paper and its Supporting Information files.

## Abstract

### Background

Preliminary empirical data indicates a substantial impact of the COVID-19 pandemic on well-being and mental health. Individuals with minoritized sexual and gender identities are at a higher risk of experiencing such negative changes in their well-being. The objective of this study was to compare levels of well-being among cis-heterosexual individuals and individuals with minoritized sexual and gender identities during the COVID-19 pandemic.

### Methods

Using data obtained in a cross-sectional online survey between April 20 to July 20, 2020 (N = 2332), we compared levels of well-being (WHO-5) across subgroups (cis-individuals with minoritized sexual identities, individuals with minoritized gender identities and cis-heterosexual individuals) applying univariate (two-sample t-test) and multivariate analysis (multivariate linear regression).

**Funding:** The study was funded by sources of the Department of Obstetrics and Gynecology and Center for Gynecological Endocrinology and Reproductive Medicine, University Hospital, LMU Munich and the Mood Disorders Research Unit of Charité – Universitätsmedizin Medicine Berlin, Department of Psychiatry and Psychotherapy, Campus Charité Berlin Mitte. We acknowledge funding by the MHB Open Access Publication Fund supported by the German Research Association (DFG).

**Competing interests:** Pichit Buspavanich received a research grant from Gilead. Sven Mahner reports Research support, advisory board, honoraria and travel expenses from AbbVie, AstraZeneca, Clovis, Eisai, GlaxoSmithKline, Medac, MSD, Novartis, Olympus, PharmaMar, Pfizer, Roche, Sensor Kinesis, Teva, and Tesaro. All other authors have no conflict of interest to declare We confirm that our Competing Interest statement does not alter the adherence to PLOS ONE policies on sharing data and materials.

**Abbreviations:** LGBTQIA, minoritized sexual and gender identities like lesbian, gay, bisexual, trans*, non-binary, queer, inter* and asexual; WHO, World Health Organization.

## Results

Results indicate overall lower levels of well-being as well as lower levels of well-being in minoritized sexual or gender identities compared to cis-heterosexual individuals. Further, multivariate analyses revealed that living in urban communities as well as being in a relationship were positively associated with higher levels of well-being. Furthermore, a moderation analysis showed that being in a relationship reduces the difference between groups in terms of well-being.

## Conclusion

Access to mental healthcare for individuals with minoritized sexual and gender identities as well as access to gender-affirming resources should be strengthened during COVID-19 pandemic. Healthcare services with low barriers of access such as telehealth and online peer support groups should be made available, especially for vulnerable groups.

## Introduction

In December 2019, the novel virus SARS-CoV-2 (COVID-19) was detected. About 13 months later more than 93 million people have been infected by COVID-19 and more than 2.5 million people died by February 2021 [1]. On 24th January 2020 the first case of COVID-19 has been reported in Germany [2]. The World Health Organization (WHO) designated COVID-19 as a Public Health Emergency of International Concern on 30th January 2020. On 11th March 2020 the WHO declared COVID-19 a pandemic. By the time of writing in April 2021, the covid-19 pandemic reached a level of humanitarian crisis. To decelerate the rapid transmission of COVID-19, many non-pharmacological interventions were enforced as a way to control the pandemic. On 22nd March the first measures of social isolation including home confinement were declared by the German federal states. Quarantine is a proven and successful measure to combat infectious diseases (e.g., Ebola); still, the global extent of confinement has never been higher [3]. Accordingly, the socioeconomic consequences of the COVID-19 pandemic in Germany have been significant. There has been an increase in unemployment, an economic recession, and an increase in social inequality, e.g. in income, education [4], and health [5]. Public support programs by the federal and state governments attempted to mitigate these consequences. Among several economic funding schemes for companies to prevent insolvency, the government offered financial aid for employees in receipt of a reduced salary, as well as financial aid for families and students in need [6–8]. COVID-19 and the containment measures have led to significant changes in certain lifestyle behaviors as well as in life satisfaction and general well-being [9–11]. Up to now, most of the research on the COVID-19 pandemic has focused on physical health. Literature shows a significant effect of the COVID-19 pandemic on daily living with an impairment in quality of life and an increase in uncertainty of the present and the future, distress, fear and panic [12]. This has negative effects on well-being and mental health with an increase of emotional disorders [11,13–15].

In general, mental distress and acute symptoms such as apprehension, stress, depression, panic or anxiety, and chronic symptoms such as insomnia and post-traumatic stress disorder (PTSD), have increased since the COVID-19 pandemic [12,16,17]. Individuals with minoritized sexual and gender identities like lesbian, gay, bisexual, trans*, non-binary, queer, inter* and asexual (LGBTQIA*) constitute a particularly vulnerable social and medical group. Past

research acknowledges that individuals with minoritized sexual and gender identities experience greater health disparities compared to heterosexual and cisgender individuals, respectively [18]; gender minority people stand out as particularly vulnerable within the LGBQTIA* group [19–21]. These health disparities are understood to be consequences of minority stress due to social disadvantages, discrimination and stigmatization in all areas of life [22,23]. Conceptual frameworks, such as the Minority Stress Model, explain the association between discrimination, social stress, and mental health among LGBTQIA* communities [24,25] The Minority Stress Model postulates that individuals belonging to minoritized groups are exposed to unique stress related to their race, gender or sexual orientation. The model includes external distal minority stressors such as discrimination or exposure to violence as well as internal proximal minority stressors such as expectations of rejection or internalized stigma. As protective factors against minority stress, the model includes social support and community connectedness [20,24,25]. Over the last years, research on needs and mental health of LGBTQIA* is increasing [26]. Overall, the incidence of mental health disparities and psychological disorders such as depression, anxiety disorders, suicidality, and substance abuse was often found to be significantly higher among LGBTQIA* individuals compared to the non-LGBTQIA* population [27–29], in particular among individuals with minoritized gender identities [19,20,30]. In Germany, the legal situation of minoritized sexual and gender identities has improved during the last years. The General Equal Treatment Act of 2006 prohibits unequal treatment based on sexual orientation and gender in civil and labor law [31], whereby trans* and inter* persons are explicitly included in the gender clause. In addition, since 2011, trans* individuals are able to change their gender registration without having undergone gender reassignment surgery involving forced sterilization. In 2017, marriage was opened to same-sex couples, also granting adoption rights to same-sex couples. Since 2018, the German state recognizes the third gender entry "diverse" in addition to "male" and "female", particularly aimed at inter* people [31]. At the same time, however, there has been an increase in anti-LGBTQIA* crime and a strengthening of right-wing populism. Corresponding political parties fight against diversity and LGBTQIA* rights and propagate traditional gender roles [32]. Crises such as the COVID-19 pandemic and resulting trauma put individuals with minoritized sexual and gender identities at a higher psychological risk of decreased well-being and exacerbation of preexisting mental health problems [33,34]. Barriers to medical care in the context of COVID-19 include, among others, a lack of access to LGBTQIA*-friendly medical care, psychotherapy and social support groups in general, and for those seeking gender-affirming healthcare and services in particular [11]. Despite the well-documented vulnerabilities of LGBTQIA* individuals, so far, there is no empirical work published that focused on a comparison of psychological needs, general well-being and risk or protective factors during the COVID-19 pandemic among people with minoritized sexual and/or gender identities and cis-heterosexual individuals. Yet, it is crucial to act immediately in response to the effects of the COVID-19 pandemic on well-being and mental health, in particular for vulnerable social groups such as individuals with minoritized sexual and gender identities. Further, it is of great interest to explore the role of general protective factors for well-being and mental health such as age [35,36], employment [35], partnership status [37,38], place of living [39,40], and children [38] during the COVID-19 pandemic.

## Aim of the present study

The present study aims to examine the impact of the COVID-19 pandemic and the precautionary social isolation measures in Germany on current well-being of individuals with minoritized sexual and gender identities (LGBTQIA*) and cis-heterosexual individuals. Thereby, particular attention is paid to heterogeneity within the larger LGBTQIA* group by comparing

cis- and trans* individuals to one another, regardless of their sexual orientation; and by drawing within-group comparisons regarding sexual orientation (i.e., comparing cis-lesbian- or gay-identifying individuals to cis-bisexual individuals). Further, general protective and risk factors associated with well-being for all individuals will be examined. Based on previous literature, the following hypothesis are proposed: Levels of well-being in the overall population in Germany have decreased during the COVID-19 pandemic compared to levels before the COVID-19 pandemic (Hypothesis 1), and levels of well-being are lower among all LGBTQIA* populations (cis-individuals with LGBQA* identities and individuals with minoritized gender identities) compared to a cis-heterosexual population (Hypothesis 2). Further, we examined whether age, residential environment, employment status, relationship status, parenthood and COVID-19 status are associated with well-being and the sexual and gender identity (Research Question). Based on our results, we derive recommendations for healthcare providers and public policy makers.

## Materials and methods

### Setting, study design and sample

An anonymous nationwide online cross-sectional survey was conducted, using SoSci Survey as a platform. The survey was administered in German language and shared via online invitations, where people were invited to partake in a survey on sexuality and family planning during the Covid-19 crisis. Between April 20 to July 20, 2020, a link with access to the survey was distributed in social communication networks on FacebookTM, InstagramTM, TwitterTM and WhatsappTM. We posted invitations to the questionnaire on national social and communication groups on FacebookTM as online bulletin boards for cities and regions in Germany (e.g., "Bulletin board Hamburg" (Schwarzes Brett Hamburg)). Further, the link to the online survey was promoted in LGBTQIA* community support groups like the FacebookTM group "Queer in Germany (LGBTQ+)" (Queer in Deutschland (LGBTQ+)). Moreover, the questionnaire was distributed within several networks of LGBTQIA* communities like the "Lesbian and Gay Federation in Germany" (Lesben- und Schwulenverband–LSVD), the largest German non-governmental LGBT rights organization. Additionally, we promoted the survey through email distribution lists of Ludwig Maximilian University of Munich. Some participants promoted the survey within their own networks (snowball sampling). Prior to data collection, all participants reviewed and accepted an online-based consent page which included information on the research project. Participation in the study was anonymous, voluntarily and without any compensation. The survey was registered by the Ethics Review Committee of the Faculty of Medicine, LMU Munich (registration number: 20-344KB) and conducted with accordance of the Declaration of Helsinki. A total of N = 2463 participants participated in the online survey. To maximize participation, the inclusion criteria were held broad and included only (1) a minimum age of 18 years and (2) German proficiency. Due to missing values, n = 131 participants had to be excluded from the analysis. Therefore, eligible participants for analysis resulted in N = 2332 participants.

### Measures

**Gender identity and sexual orientation.** Gender identity and sexual orientation were assessed with the item "*In your opinion, which of the following categories most apply to you*?". The following answer categories were provided: *heterosexual, homosexual, bisexual, asexual, female, male, cis* ("*I identify with the gender assigned at birth*"), *trans** ("*I do not identify with the gender assigned at birth*") and *others*. Note that we are aware of the pathologizing nature of the term "homosexual". In Germany, the term "*homosexuell*" is still widely used; for reasons of

transparency, we report the direct translation as it appeared in the survey question. Multiple answers were possible. For the purpose of the analysis, we divided all participants into 12 groups according to their self-assigned gender identity and sexual orientation: (I) cis-hetero-sexual women, (II) cis-heterosexual men, (III) cis-lesbian women, (IV) cis-gay men, (V) cis-bisexual women, (VI) cis-bisexual men, (VII) cis-asexual women, (VIII) cis-asexual men, (IX) trans* women, (X) trans* men, (XI) non-binary gender identities (participants who identify as female and male), and (XII) inter* people. The star (*) indicates that the respective terms include further gender identities beyond the expression transgender, transident, transsexual and inter*, respectively. Regardless of their sexual orientation, all participants who reported a minoritized gender identity to their form one analytical group separate from cis-gendered people with minoritized sexual identities in the presented descriptive statistics and the analyses. The three analytical groups are: *individuals with minoritized sexual identities* (cis-lesbian women, cis-gay men, cis-bisexual women, cis-bisexual men, cis-asexual women and cis-asexual men), *individuals with minoritized gender identities* (trans* women, trans* men, non-binary gender and inter* people) and *cis-heterosexual individuals* (cis-heterosexual women and cis-heterosexual men). For the multivariate analysis we used dichotomous variables for cis-lesbian/gay, cis-bisexual, cis-asexual, trans*, non-binary and inter* individuals whereby cis-heterosexual served as the reference group in each of these variables.

**Well-being.**   We used the 5-item short version of World Health Organization-Five Well-Being Index-10 (WHO-5) to measure current mental well-being [41]. The WHO-5 is a brief self-reported questionnaire, which consists of five items assessing subjective psychological well-being over a 14-day period. Each item is scored from 0 (*none of the time*) to 5 (*all of the time*). The total raw score ranges from 0 to 25, whereby higher values indicate better well-being. A total raw score $\leq 13$ indicates a clinically significant depression [6]. The final score is calculated multiplying the total raw score by 4, with 0 representing the worst imaginable well-being and 100 representing the best imaginable well-being. This was conducted in order to compare the values with data from the German validation study of the WHO-5 [42]. In the present study, the scale showed very good internal consistency (Cronbach's alpha = 0.873).

**Protective factors for well-being.**   Items were treated as categorical variables. For more details, see descriptive statistics in Table 1. In the multivariate analysis we used the following dichotomous variables: *age* (under 35 years versus 36 years and above), *employment status* (employed versus not employed (including students)), *residential environment* (urban cities versus rural communities under 20,000 inhabitants), *relationship status* (single versus in a relationship), *parenthood* (yes/no), and *COVID-19 status* (current or previous COVID-19 infection versus not infected or not tested).

## Statistical analyses

This study focuses on the impact of the COVID-19 precautionary measures on well-being in individuals with minoritized sexual and gender identities compared to cis-heterosexual individuals. First, descriptive statistics were calculated for all subgroups and variables of interest (Table 1). To explore Hypothesis 1, assuming that individual's current overall levels of well-being in Germany have decreased during the COVID-19 pandemic compared to prior levels before the COVID-19 pandemic, t-tests against a fixed value from a previous study [42] were conducted. To test Hypothesis 2, saying that individual's current overall levels of well-being are lower among LGBTQIA* populations compared to a cis-heterosexual population, we conducted a two-sample t-test for the comparison of the groups cis-heterosexual vs. not cis-het-erosexual as well as an ANOVA with post-hoc tests for more differentiated insights. To analyze which group has a higher probability of being below the WHO-5 cut-off score, we

**Table 1. Descriptive statistics of the sample; n (%).**

| N = 2332 | Cis-heterosexual individuals | | Cis-individuals with minoritized sexual identities | | | | | | Individuals with minoritized gender identities | | | |
|---|---|---|---|---|---|---|---|---|---|---|---|---|
| | Women | Men | Lesbian women | Gay men | Bisexual women | Bisexual men | Asexual women | Asexual men | Trans* women | Trans* men | Non-binary | Inter* |
| | n = 1 004 | n = 300 | n = 353 | n = 108 | n = 254 | n = 80 | n = 29 | n = 7 | n = 60 | n = 96 | n = 31 | n = 10 |
| **Age** | | | | | | | | | | | | |
| 18–25 years | 389 (38,8) | 123 (41.0) | 90 (25.5) | 18 (16.7) | 95 (37.4) | 25 (31.3) | 14 (48.3) | 3 (42.9) | 12 (20.0) | 27 (28.1) | 9 (30.0) | 6 (60.0) |
| 26–35 years | 476 (47.5) | 118 (39.9) | 141 (39.9) | 50 (46.3) | 102 (40.2) | 24 (30.0) | 12 (41.4) | 2 (28.6) | 34 (56.7) | 30 (31.3) | 11(36.7) | 1 (10.0) |
| 36–45 years | 116 (11.6) | 35 (11.7) | 93 (26.3) | 19 (17.6) | 46 (18.1) | 26 (32.5) | 0 (0.0) | 2 (28.6) | 12 (20.0) | 36 (37.5) | 8 (26.7) | 0 (0.0) |
| Over 46 years | 21 (2.1) | 24 (8.0) | 29 (8.2) | 21 (19.4) | 11 (4.3) | 5 (6.3) | 3 (10.3) | 0 (0.0) | 2 (3.3) | 3 (3.1) | 2 (6.7) | 3 (30.0) |
| **Relationship status** | | | | | | | | | | | | |
| Single | 267 (26.6) | 102 (34.0) | 81 (22.9) | 33 (30.6) | 100 (39.4) | 43 (53.8) | 19 (65.5) | 6 (85.7) | 28 (46.7) | 58 (60.4) | 16 (51.6) | 8 (80.0) |
| In a relationship | 737 (73.4) | 198 (66.0) | 272 (77.1) | 75 (69.4) | 154 (60.6) | 37 (46.3) | 10 (34.5) | 1 (14.3) | 32 (53.3) | 38 (39.6) | 15 (48.4) | 2 (20.0) |
| **Parenthood** | | | | | | | | | | | | |
| yes | 823 (87.6) | 223 (82.0) | 250 (77.2) | 87 (87.0) | 202 (84.2) | 44 (62.0) | 24 (85.7) | 5 (100.0) | 52 (86.7) | 82 (88.2) | 21 (84.0) | 6 (75.0) |
| no | 117 (12.4) | 49 (18.0) | 74 (22.8) | 13 (13.0) | 38 (15.8) | 27 (38.0) | 4 (14.3) | 0 (0.0) | 8 (13.3) | 11 (11.8) | 4 (16.0) | 2 (25.0) |
| **Residential environment** | | | | | | | | | | | | |
| Metropolis[1] | 674 (67.3) | 184 (61.3) | 178 (50.6) | 67 (62.0) | 153 (60.5) | 58 (72.5) | 18 (62.1) | 6 (85.7) | 46 (76.7) | 59 (61.5) | 25 (80.6) | 8 (80.0) |
| Medium-sized town[2] | 131 (13.1) | 46 (15.3) | 73 (20.7) | 12 (11.1) | 34 (13.4) | 7 (8.8) | 6 (20.7) | 0 (0.0) | 10 (16.7) | 12 (12.5) | 4 (12.9) | 0 (0.0) |
| Small town[3] | 107 (10.7) | 40 (13.3) | 44 (12.5) | 23 (21.3) | 31 (12.3) | 10 (12.5) | 3(10.3) | 0 (0.0) | 21 (21.9) | 2 (6.5) | 1 (10.0) | 1 (10.0) |
| Rural community[4] | 90 (9.0) | 30 (10.0) | 57 (16.2) | 6 (5.6) | 35 (13.8) | 5 (6.3) | 2 (6.9) | 1 (14.3) | 1 (1.7) | 4 (4.2) | 0 (0.0) | 1 (10.0) |
| **Employment status** | | | | | | | | | | | | |
| Self-employed | 53 (5.3) | 11 (3.7) | 23 (6.5) | 6 (5.6) | 14 (5.5) | 4 (5.0) | 5 (17.2) | 0 (0.0) | 4 (6.7) | 5 (5.3) | 2 (6.5) | 0 (0.0) |
| Employed | 464 (46.3) | 135 (45.2) | 244 (69.3) | 80 (74.8) | 135 (53.1) | 49 (61.3) | 8 (27.6) | 3 (42.9) | 38 (63.3) | 52 (54.7) | 17 (54.8) | 2 (20.0) |
| Student | 425 (42.4) | 136 (45.5) | 59 (16.8) | 18 (16.8) | 82 (32.3) | 23 (28.7) | 14 (48.3) | 4 (57.1) | 7 (11.7) | 18 (18.9) | 7 (22.6) | 7 (70.0) |
| Not employed | 60 (6.0) | 17 (5.7) | 26 (7.4) | 3 (2.8) | 23 (9.1) | 4 (5.0) | 2 (6.9) | 0 (0.0) | 11 (18.3) | 20 (21.1) | 5 (16.1) | 1 (10.0) |
| **COVID-19 status** | | | | | | | | | | | | |
| Infected, symptoms | 0 (0.0) | 2 (0.7) | 0 (0.0) | 1 (1.0) | 0 (0.0) | 0 (0.0) | 0 (0.0) | 0 (0.0) | 0 (0.0) | 0 (0.0) | 0 (0.0) | 0 (0.0) |
| Infected, no symptoms | 2 (0.2) | 0 (0.0) | 1 (0.3) | 1 (1.0) | 0 (0.0) | 1 (1.3) | 0 (0.0) | 0 (0.0) | 0 (0.0) | 0 (0.0) | 0 (0.0) | 0 (0.0) |
| Previous infected | 7 (0.7) | 2 (0.7) | 2 (0.6) | 0 (0.0) | 0 (0.0) | 1 (1.3) | 0 (0.0) | 1 (16.7) | 0 (0.0) | 0 (0.0) | 0 (0.0) | 0 (0.0) |
| Not infected/not tested | 983 (99.1) | 294 (98.7) | 337 (99.1) | 101 (98.1) | 250 (100) | 74 (97.4) | 29 (100) | 5 (83.3) | 58 (100) | 96 (100) | 30 (100) | 10 (100) |

*Notes*: [1] = 100,000 or more inhabitants,

[2] = 20,000 to 100,000 inhabitants,

[3] = 5,000 to 20,000 inhabitants,

[4] = up to 5,000 inhabitants.

performed a logistic regression. To answer the Research Question, we examined the association between subgroups and current well-being by conducting a multivariate linear regression. We used the WHO-5 final score as our dependent variable, and the dummy variables of the individual identities (cis-heterosexual as reference versus cis-lesbian/gay, cis-bisexual, cis-asexual, trans*, non-binary and inter*) as independent variables. Further, we included the following protective factors for well-being in the model: age, residential environment, employment status, relationship status, parenthood and COVID-19 status. Hayes' PROCESS tool (model 1) was used for moderation analyses to test the influence of potentially protective factors. A significance level of 0.05 was set for all analyses. We used SPSS (Version 26) and RStudio (Version 1.4.1106) for statistical analysis.

## Results

In total, N = 2332 participants were included in the analysis. Of those, n = 1304 (55.9%) self-identified as *cis-heterosexual individuals*, n = 832 (35.6%) self-identified as *individuals with minoritized sexual identities*, and n = 197 (8.4%) self-identified as *individuals with minoritized gender identities*. Among the cis-gender respondents, there were more women (n = 1640, 76.8%) than men (n = 495, 23.2%). Concerning age, the vast majority of individuals with minoritized sexual identities, individuals with minoritized gender identities and cis-heterosexual individuals were younger than 36 years old. In terms of employment status most of the participants across all three groups n = 1354 (58.1%) were currently working and a total of n = 800 (34.3%) participants were students. Regarding the residential environment, most participants n = 1811 (77.7%) lived in urban cities with more than 20,000 inhabitants. Only n = 8 participants reported a current COVID-19 infection and n = 13 reported a previous COVID-19 infection. For more details on each individual group, descriptive statistics of the sample can be obtained from Table 1.

To test Hypothesis 1, the mean value of well-being M = 75.6 (SD = 13.85) from the study by Brähler et al. [42] was used as a fixed value. Results have shown that levels of well-being in all groups were significantly lower compared to the mean score from the reference sample by Brähler and colleagues. However, it is important to note that the standard deviations of the current study were higher than in the reference survey sample (for more detail see Table 2).

To test Hypothesis 2, a two sample t-test revealed that participants within the cis-heterosexual group had a significant higher mean level of well-being (M = 54.90, SD = 20.20) compared to participants who assigned themselves to the overall LGBTQIA* group (M = 51.31, SD = 21.09), t(1711.40) = 3.89, p = < 0.001, d = 0.17. ANOVA results showed a significant group effect, F(6,2325) = 7.51, p < 0.001 indicating differences in well-being between the subgroups within the LGBTQIA* group. Post-hoc tests were calculated to determine which groups differ significantly in well-being from the cis-heterosexual group. Results revealed that participants from the group cis-bisexual (p < 0.001), cis-asexual (p = 0.024) and trans* (p = 0.001) individuals showed significantly lower levels of well-being compared to participants from the cis-heterosexual group. In terms of clinically significant depression, descriptive analyses revealed that bisexual individuals reported the highest levels of clinical depression on the WHO-5, followed by cis-asexual individuals, non-binary individuals and trans* individuals. The lowest level of clinically significant depression was reported by cis-heterosexual individuals and cis-lesbian/gay individuals (for more detail see Table 2).

A chi-squared test indicated that the overall effect of the group is statistically significant (see Table 3). A logistic regression revealed that for cis-heterosexual individuals (reference), the log odds of being below the cut-off value (versus above) decrease significantly. However, for bisexual as well as for trans* individuals the log odds of being below the cut-off value (versus above)

**Table 2. Means, standard deviations and ranges for the WHO-5 across individuals and subgroups.**

| Group | N | M[1] | SD[2] | Range[3] | Cut-off[7]n (%) | t(df)[8] | p-value[8] | d[9] |
|---|---|---|---|---|---|---|---|---|
| *Individuals* | | | | | | | | |
| Cis-heterosexual | 1 304 | 54.90 | 20.20 | 0–100 | 586 (44.9) | -37.02 (1303) | < **0.001** | 1.20 |
| Women | 1 004 | 55.02 | 20.02 | 0–100 | 447 (44.5) | | | |
| Men | 300 | 54.50 | 20.82 | 0–100 | 139 (46.3) | | | |
| Cis-lesbian/gay | 461 | 54.08 | 20.70 | 0–100 | 200 (43.4) | -22.32 (460) | < **0.001** | 1.22 |
| Women | 353 | 53.81 | 20.39 | 4–96 | 158 (44.8) | | | |
| Men | 108 | 54.96 | 21.74 | 0–100 | 42 (38.9) | | | |
| Cis-bisexual | 334 | 47.95 | 20.86 | 0–100 | 208 (62.3) | -24.22 (333) | < **0.001** | 1.56 |
| Women | 254 | 49.31 | 20.87 | 0–100 | 151 (59.4) | | | |
| Men | 80 | 43.65 | 20.38 | 0–88 | 57 (71.3) | | | |
| Cis-asexual | 36 | 47.00 | 23.39 | 4–96 | 20 (55.6) | -7.34 (35) | < **0.001** | 1.49 |
| Women | 29 | 42.35 | 21.42 | 4–80 | 18 (61.1) | | | |
| Men | 7 | 66.29 | 22.61 | 40–96 | 2 (28.6) | | | |
| Trans* | 156 | 49.00 | 23.89 | 4–92 | 84 (53.8) | -13.91 (155) | < **0.001** | 1.36 |
| Women | 60 | 50.08 | 23.66 | 12–92 | 29 (48.3) | | | |
| Men | 96 | 47.88 | 24.09 | 4–84 | 55 (57.3) | | | |
| Non-binary | 31 | 48.26 | 23.19 | 16–80 | 17 (54.8) | -6.57 (30) | < **0.001** | 1.43 |
| Inter* | 10 | 64.00 | 13.60 | 44–76 | 4 (40.0) | -2.70 (9) | 0.024 | 0.85 |
| *Total sample/Subgroups* | | | | | | | | |
| Total sample | 2 332 | 53.18 | 20.90 | 0–100 | 1 119 (48.0) | -51.81 (2331) | < **0.001** | 1.37 |
| Minoritized sexual identities | 831 | 51.29 | 21.09 | 0–100 | 428 (51.5) | -33.20 (830) | < **0.001** | 1.36 |
| Minoritized gender identities | 197 | 49.64 | 23.52 | 4–92 | 105 (53.3) | -15,48 (196) | < **0.001** | 1.35 |

*Notes*: [1] = mean of the well-being final score,

[2] = standard deviation of the well-being final score,

[3] = range of the well-being final score,

[7] = total raw score ≤ 13 indicates a clinically significant depression,

[8] one sample t-tests against the value M = 75.6 (mean well-being final score in the reference sample from Brähler et al. (2007)). Please note that the sample of Brähler et al. consisted of mainly cis-heterosexual individuals.

[9] = Cohen's d.

increase compared to the reference group. All other estimates were not significant. To answer the Research Question, a multivariate linear regression model (see Table 3) with the WHO-5

**Table 3. Logistic regression of self-assigned sexual and gender identities on clinically significant depression as dependent variable measured with WHO-5[1].**

| | Chi[2] (df) | Regression coefficient | Standard Error | Odds Ratio | p-value | 95%-confidence interval | |
|---|---|---|---|---|---|---|---|
| | | | | | | Lower | Upper |
| Group | 39.90 (6) | | | | < **0.001** | | |
| Intercept (Cis-heterosexual as reference) | | -0.20 | 0.06 | 0.81 | < **0.001** | 0.73 | 0.91 |
| Cis-lesbian/gay | | -0.06 | 0.11 | 0.93 | 0.56 | 0.75 | 1.16 |
| Cis-bisexual | | 0.70 | 0.13 | 2.02 | < **0.001** | 1.58 | 2.59 |
| Cis-asexual | | 0.42 | 0.34 | 1.53 | 0.21 | 0.78 | 3.02 |
| Trans* | | 0.35 | 0.17 | 1.42 | **0.04** | 1.02 | 1.99 |
| Non-binary | | 0.39 | 0.37 | 1.48 | 0.28 | 0.72 | 3.09 |
| Inter* | | -0.20 | 0.65 | 0.81 | 0.75 | 0.20 | 2.87 |

Note: [1] = 5-item short version of World Health Organization-Five Well-Being Index-10.

final score as the dependent variable revealed that there is a significant negative association between the level of well-being and minoritized sexual identities; in particular cis-asexual individuals, followed by cis-bisexual individuals. Minoritized gender identities, in particular non-binary and trans* individuals, were also negatively associated with well-being. Living in urban communities and being in a relationship were positively association with the level of well-being in this model. All other covariates had no significant effect.

Since the multivariate linear regression analysis showed that the variables relationship status and residential environment have a significant influence on well-being, we explored in a next step whether these variables influence the effect of individual identity on well-being. To test whether relationship status influences the association between minoritized sexual and gender identities (cis-heterosexual vs. LGBTQIA*) and well-being, a moderation analysis was conducted. Results showed a significant positive interaction effect of LGBTQIA* identities and relationship status (see Table 4). Substantively this means, as revealed by the conditional effects, that LGBTQIA* individuals without a partner have particularly low well-being, whereas the gap to cis-heterosexual individuals is narrower for LGBTQIA* individuals with a partner. In other words, the results indicate that the negative association between well-being and LGBTQIA* identities is mitigated by being in a relationship but not erased completely. A further analysis concerning the residential environment showed no moderator effect (see Table 4).

## Discussion

The present study provides unique evidence on the comparison of well-being among cis-heterosexual individuals and LGBTQIA* individuals during the COVID-19 pandemic. Based on previous literature, we hypothesized overall lower levels of well-being in all individuals during the COVID-19 pandemic compared to levels prior to the COVID-19 pandemic (based on prior empirical work; Hypothesis 1); as well as lower levels of well-being among individuals with minoritized gender and/or sexual identities compared to cis-heterosexual individuals (Hypothesis 2). Further, the influence of age, residential environment, employment status, relationship status, parenthood and COVID-19 status in the association between well-being and sexual and gender identities was explored (Research Question).

First, overall results of the present study confirmed lower levels of well-being in all groups during the COVID-19 pandemic. Comparing these results with data obtained during the validation of the German version of the WHO-5 prior to the COVID-19 pandemic, we found a significant lower overall mean of the well-being score [42]. This finding is in line with our Hypothesis 1 as lower levels of well-being were expected during the current COVID-19 pandemic. A recently published empirical study reporting on mental health during the COVID-19 pandemic in Germany has similarly found an overall decrease in well-being measured with the WHO-5 [43]. In light of staggering preexisting mental health gradients between LGBTQIA* and cis-heterosexual people in Germany–documented just prior to the pandemic [21]–the findings of the current study are alarming.

Second, when comparing well-being levels among all individuals with minoritized gender and sexual identities with cis-heterosexual individuals, results indicated higher levels of well-being among cis-heterosexual individuals compared to LGBTQIA* people. The results were mainly driven by the group of cis-bisexual, cis-asexual and trans* individuals, which showed significant lower levels in well-being compared to participants from the cis-heterosexual group. The finding concerning LGBTQIA* individuals as one group was expected (Hypothesis 2) and is in line with previous empirical work [21]. Research in this field is growing, and the vast majority of studies report significantly poorer well-being and mental health in individuals with minoritized sexual and/or gender identities when compared with heterosexual and

**Table 4. Linear regression model with WHO-5 final score as dependent variable and moderation analyses.**

| | Regression coefficient | Standardized coefficient | Standard error | p-value | 95%-confidence interval | |
|---|---|---|---|---|---|---|
| | | | | | Lower | Upper |
| Intercept | 47.95 | | 10.30 | < **0.001** | 27.76 | 68.14 |
| Individual identities | | | | | | |
| - Cis-heterosexual (reference) | - | | - | - | - | - |
| - Cis-lesbian/gay | -0.67 | -0.01 | 1.22 | 0.59 | -3.04 | 1.73 |
| - Cis-bisexual | -6.90 | -0.12 | 1.34 | < **0.001** | -9.51 | -4.26 |
| - Cis-asexual | -8.80 | -0.05 | 3.66 | **0.02** | -15.98 | -1.62 |
| - Trans* | -4.56 | -0.06 | 1.82 | **0.01** | -8.12 | -0.10 |
| - Non-binary | -9.73 | -0.05 | 4.36 | **0.03** | -18.27 | -1.19 |
| - Inter* | 11.85 | 0,04 | 7.35 | 0.10 | -2.56 | 26.26 |
| Age[a] | -0.77 | -0.02 | 1.19 | 0.51 | -3.09 | 1.55 |
| Relationship status[b] | 3.02 | 0.07 | 0.99 | < **0.001** | 1.07 | 4.97 |
| Employment status[c] | -1.50 | -0.04 | 1.12 | 0.18 | -3.69 | 0.68 |
| Residential enviroment[d] | -1.92 | -0.04 | 0.96 | **0.046** | -3.79 | -0.04 |
| Parenthood[e] | -2.21 | -0.00 | 1.30 | 0.09 | -4.76 | 0.34 |
| Covid-19 status[f] | 5.31 | 0.24 | 4.76 | 0.27 | -4.03 | 14.65 |
| **Statistics for linear regression** | | | | | | |
| $R^2$ | 0.03 | | | | | |
| Adjusted $R^2$ | 0.03 | | | | | |
| Standard Error | 20.62 | | | | | |
| F-statistic | 5.60** | | | | | |
| **Moderation Analyses[g]** | | | | | | |
| Identity[h]* Relationship status[b] | 3.98 | 0.20 | 1.95 | **0.04** | 0.15 | 7.81 |
| Identity[h] without a partner | -6.05 | -0.13 | 1.09 | < **0.001** | -9.22 | -2.89 |
| Identity[h] in a relationship | -2.07 | -0.05 | 1.09 | 0.06 | -4.22 | 0.08 |
| Identity[h]*Residential enviroment[d] | 0.91 | 0.04 | 0.18 | 0.63 | -2.76 | 4.60 |

*Notes*: a. 1 = under 35 years; 2 = 36 years and above;

b. 1 = single, 2 = in a relationship;

c. 1 = employed, 2 = not employed including students;

d. 1 = urban cities, 2 = rural communities under 20.000 inhabitants;

e. 1 = no children, 2 = one or more children;

f. 1 = current or previous COVID-19 infection, 2 = not infected or not tested;

g. using Hayes' PROCSS tool (model 1);

h. 1 = cis-heterosexual individuals, 2 = individuals with minoritized sexual and/or gender (LGBTQIA*) identities.

cisgender individuals [44–47]. A recent systematic literature review of N = 77 studies reported higher general distress, depressive symptoms, anxiety, suicidality as well as exposure to trauma and substance abuse among LGBTQIA* individuals [20]. Further, in line with the Minority Stress Model [24,25], EU-LQBTI surveys reported high levels of discrimination in access to and experience with healthcare [48,49]. LGBTQIA* individuals experience numerous health disparities and poor mental health, especially in times of the COVID-19 pandemic [50]. Another study where mental health of LGBT college students from the U.S. during the COVID-19 pandemic was examined, reported that approximately 60% of the sample were experiencing psychological distress, anxiety, and depression during the pandemic [34].

Our analyses further showed well-being heterogeneity within the LGBTQIA* group. With regard to sexual identity among cis-individuals, results suggest asexual and bisexual

individuals at particular risk for poor mental health. In addition, the results show that the likelihood of being below the cut-off value for clinically significant depression was significantly increased in bisexual individuals and significantly decreased in heterosexual individuals. This finding is in line with previous empirical work [51–53]. For example, in a systematic review and meta-analysis on the prevalence of depression and anxiety, higher rates of depression and anxiety were consistently reported among bisexual individuals compared to heterosexual individuals and higher or equivalent rates in comparison to lesbian/gay individuals [54].

Notably, we did not find a significant difference between cis-lesbian/gay and cis-heterosexual individuals in regards to clinically significant depression in our data, nor does the effect suggest any sizable difference in means. Existing literature would suggest that lesbian women and gay men have poorer mental health than heterosexual individuals. In light of the mounting evidence that such a well-being gap exists, self-selection into the current study may play a role; specifically, particularly well-adjusted cis-lesbian women and cis-gay men or cis-heterosexual people with particularly poor mental health who dampen the group difference in well-being. Indeed, the sample is made up of many cis-women, whose consistently higher levels of depression than men are well-documented [55]. The sample further contains a noticeably large share of cis-lesbian and cis-gay individuals who are in relationships; also possibly a by-product of the family planning topic. As we have illustrated in this study, having a partner is a factor that minimizes the adverse well-being outcomes, which is another reason that likely led us to underestimate the well-being gap between cis-lesbian/gay individuals compared to cis-heterosexual individuals. In other words, the overall difference we found in this study between cis-heterosexual and LGBTQI* individuals is likely even more pronounced in general populations with adequate representation of cis men in general and unpartnered cis-gay/lesbian identifying individuals.

Regarding the analyses of individuals with minoritized gender identities, heightened rates of clinical depression among trans* and non-binary people were found in the regression analyses. These findings mirror the existing literature on mental health disparities of trans* and gender non-conforming people. For example, a recent study confirmed that due to the COVID-19 pandemic, access to gender-affirming resources and the ability of transgender and non-binary people to live according to their preferred gender has been reduced [56]. While gender-affirming care has repeatedly been found to improve physical and mental health of people with minoritized gender identities [57,58], access to it was already difficult for many individuals. The COVID-19 pandemic created an additional burden. Negative impacts of the pandemic include deferrals of and limited access to gender-affirming treatment (e.g., hormonal treatment, surgery), services (e.g., hair removal, binders) as well as access to mental counseling and psychotherapy, which may be linked to increased depressive symptoms [56]. Further, similar to our results, the same study reported about half of the participants screened positive for clinically significant depression.

With regard to protective factors for well-being, results revealed that being in a partnership and living in urban areas are relevant for well-being for all individuals in general, and particularly so for LGBTQIA* individuals. Interestingly, these significant protective factors for well-being, can also be interpreted as protective factors within the Minority Stress Model [20,25]. The protective role of being in a romantic partnership has already been reported in the past [35,37]. Further, previous literature has acknowledged social support [59–61] and in particular romantic relationships [58,62], to be associated with higher levels of wellbeing and mental health among individuals with minoritized gender and sexual. In addition, results of the present study indicate the negative association between LGBTQIA* identities and well-being is mitigated by being in a relationship. Future research should focus more closely on examining the buffering role of romantic relationships in the association between individuals with

minoritized gender and sexual identities and well-being. With regard to living in urban areas, past research has repeatedly found that individuals of LGBTQIA* communities from rural areas experience high negative mental health consequences of minority stress [63]. Further, rural residents in general, but especially individuals from LGBTQIA* communities, report barriers and difficulties in the access of mental healthcare, among others because mental health services are in short supply [64]. The current COVID-19 pandemic and its restrictions represent an additional burden to healthcare access, especially for LGBTQIA* communities.

To sum up, results of the present study underline the particular vulnerability of LGBTQIA* individuals during the COVID-19 pandemic. In order to improve well-being and mental health among this group, access to mental healthcare should be strengthened during the COVID-19 pandemic, especially in rural areas. Healthcare services with low barriers of access such as telehealth, online care programs, counseling and supervision, as well as training and psychoeducation through online platforms should be made readily available. Telehealth has been found to be effective and practically feasible for the provision of mental health service during this pandemic [65]. Another recent study concluded that telehealth not only constitutes an effective health service but also has the potential of rapid implementation in both metropolitan and rural areas. Thus, access to telehealth during pandemics should be facilitated, particularly to vulnerable communities at higher risk of poor mental health. Moreover, access to gender-affirming services should not be disrupted in this time but rather strengthened. This study has substantial strengths including the use of a large and nationwide sample with high participation rates of LGBTQIA* individuals. It is the first study of its kind which examined well-being among LGBTQIA* individuals compared to cis-heterosexual individuals during the COVID-19 pandemic. However, the following limitations exist. First, well-being was measured by self-report. Self-reports of mental health may differ from clinical diagnoses. For instance, Grobe et al. [66] found administrative diagnoses within the healthcare system showed higher rates compared with survey self-reported depressive symptoms. However, self-perceptions of mental health have their own right as they are related to mental components of subjective well-being. Future research should incorporate clinical diagnosis or administrative data of mental health problems in order to assess mental health more objectively. Secondly, in the current study, LGBTQIA* individuals–in particular cis-lesbian and cis-gay identifying individuals–who are in relationships are overrepresented. The detected differences in well-being are likely more pronounced in the general population of LGBTQI* people than the current study shows. Third, while this is the first study to examine well-being among LGBTQIA* individuals compared to cis-heterosexual individuals during the COVID-19 pandemic, data were obtained at a single time point. Therefore, no trends or within-person comparisons of mental health before and after the COVID-19 pandemic can be drawn. A larger number of observations of trans* and gender non-conforming individuals would allow for further distinction of this group, according to their sexual orientation, which may interact with minoritized gender identities in unique ways. Further, as the present study is a cross sectional study only, associations and not causations can be inferred from the data. Future research on well-being of individuals with minoritized sexual and gender identities during pandemics should focus on longitudinal research.

## Conclusion

The present study's findings reveal lower levels of well-being among all participants compared to research conducted before the COVID-19 pandemic. Levels of well-being were lower among individuals with minoritized sexual and gender identities compared to cis-heterosexual individuals. Further, results indicate a protective role of being in a partnership and living in an

urban area. Access to mental healthcare and gender-affirming resources for LGBTQIA* individuals should be strengthened during the COVID-19 pandemic. Healthcare services with low barriers of access such as telehealth and online peer support groups should be made readily available, especially for vulnerable groups.

## Supporting information

**S1 Dataset.**
(SAV)

## Acknowledgments

We would like to thank Grace O'Malley (Department of Paediatric Oncology/Haematology, Charité – Universitätsmedizin Berlin, Campus Virchow-Klinikum, Berlin, Germany) for proofreading the manuscript.

## Author Contributions

**Conceptualization:** Pichit Buspavanich, Sonia Lech, Eva Lermer, Mirjam Fischer, Maximilian Berger, Theresa Vilsmaier, Till Kaltofen, Simon Keckstein, Sven Mahner, Joachim Behr, Christian J. Thaler, Falk Batz.

**Formal analysis:** Pichit Buspavanich, Sonia Lech, Eva Lermer, Falk Batz.

**Project administration:** Pichit Buspavanich, Falk Batz.

**Supervision:** Pichit Buspavanich, Falk Batz.

**Writing – original draft:** Pichit Buspavanich, Sonia Lech, Falk Batz.

**Writing – review & editing:** Pichit Buspavanich, Sonia Lech, Eva Lermer, Mirjam Fischer, Maximilian Berger, Theresa Vilsmaier, Till Kaltofen, Simon Keckstein, Sven Mahner, Joachim Behr, Christian J. Thaler, Falk Batz.

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
