## [Decision Letter · Decision Letter 0]

30 Mar 2021

PONE-D-21-06928

Well-being during COVID-19 pandemic: A comparison of individuals with minoritized sexual and gender identities and cis-heterosexual individuals

PLOS ONE

Dear Dr. Buspavanich,

Thank you for submitting your manuscript to PLOS ONE. After careful consideration, we feel that it has merit but does not fully meet PLOS ONE’s publication criteria as it currently stands. Therefore, we invite you to submit a revised version of the manuscript that addresses the points raised during the review process.

The two Reviewers made two conflicting judgments. In my opinion, I believe that the breadth of the sample collected and the correctness of the experimental design merit giving the authors a chance to respond to the Reviewers and revise the manuscript accordingly. I therefore urge the authors to resolve all comments from the two Reviewers. In particular, I urge the authors to pay particular attention to the issues raised by Reviewer 2 related to methodology (recruitment information) and proper interpretation of the data (e.g., between lesbian/gay and heterosexual individuals). Finally, I recall the importance of what Reviewer 1 raised regarding the use of a standard English language, which is a prerequisite for publication. I suggest that authors either state that the language review was conducted by a native English-speaking author or that they use a professional proofreading service.

We look forward to receiving your revised manuscript.

Kind regards,

Stefano Federici, Ph.D.

Academic Editor

PLOS ONE

Journal Requirements:

'Pichit Buspavanich received a research grant from Gilead. All other authors have no conflict of interest to declare. Sven Mahner reports Research support, advisory board, honoraria and travel expenses from AbbVie, AstraZeneca, Clovis, Eisai, GlaxoSmithKline, Medac, MSD, Novartis, Olympus, PharmaMar, Pfizer, Roche, Sensor Kinesis, Teva, and Tesaro.'

a.Please confirm that this does not alter your adherence to all PLOS ONE policies on sharing data and materials, by including the following statement: "This does not alter our adherence to  PLOS ONE policies on sharing data and materials.” (as detailed online in our guide for authors http://journals.plos.org/plosone/s/competing-interests).  If there are restrictions on sharing of data and/or materials, please state these. Please note that we cannot proceed with consideration of your article until this information has been declared.

Additional Editor Comments:

The two Reviewers made two conflicting judgments. In my opinion, I believe that the breadth of the sample collected and the correctness of the experimental design merit giving the authors a chance to respond to the Reviewers and revise the manuscript accordingly. I therefore urge the authors to resolve all comments from the two Reviewers. In particular, I urge the authors to pay particular attention to the issues raised by Reviewer 2 related to methodology (recruitment information) and proper interpretation of the data (e.g., between lesbian/gay and heterosexual individuals). Finally, I recall the importance of what Reviewer 1 raised regarding the use of a standard English language, which is a prerequisite for publication. I suggest that authors either state that the language review was conducted by a native English-speaking author or that they use a professional proofreading service.

Reviewers' comments:

Reviewer's Responses to Questions

**Comments to the Author**

1. Is the manuscript technically sound, and do the data support the conclusions?

Reviewer #1: Yes

Reviewer #2: No

2. Has the statistical analysis been performed appropriately and rigorously? 

Reviewer #1: Yes

Reviewer #2: Yes

3. Have the authors made all data underlying the findings in their manuscript fully available?

Reviewer #1: Yes

Reviewer #2: Yes

4. Is the manuscript presented in an intelligible fashion and written in standard English?

Reviewer #1: Yes

Reviewer #2: Yes

5. Review Comments to the Author

Reviewer #1: The manuscript makes an incremental but important contribution to understanding mental health during the COVID pandemic. I have two suggestions for the authors. First, a short section on the research context would be useful to readers unfamiliar with Germany. For example, details on the German social safety net and how this may have offset some of the ill effects of the pandemic, such as unemployment, are things that may have a bearing on mental health status. If the German government had any responses to help mitigate the effects of the pandemic would also be relevant. Additionally, details on the legal and social climate toward gender and sexual minorities would be useful for understanding the mental health of this population. Such details about the local context could help show to what extent the results are generalizable in other locations. Second, the manuscript would benefit from a couple of rounds of careful editing. I noticed a number of grammatical errors and awkwardly worded passages that could use some cleaning up.

Reviewer #2: This manuscript addresses differences in well-being and depression between (1) sexual and gender minority individuals (SOGI) and their (2) cisgender heterosexual peers during the Covid-19 pandemic, using a German sample recruited from social media platforms. There are theoretical, methodological, and validity concerns that dampen enthusiasm for the manuscript.

The manuscript would benefit from stronger grounding in minority stress theory. This would allow the authors, for example, to remove stress from health outcomes (p.4), to distinguish between protective factors and covariates, and to specify hypotheses pertaining to protective factors. The authors imply that SOGI individuals will be more stressed by the pandemic and, thus, experience even poorer mental health than before the pandemic. This is not testable, given their data. The authors sometimes conflate findings and issues pertaining to gender minority individuals with those of sexual minority individuals, a problem apparent in the Introduction and Discussion.

The Method section would benefit from more recruitment information. As it reads, someone could not replicate the study. For examples, what chats were accessed on the social media sites and how was the study advertised to prospective participants. Such information may explain why the authors have more lesbians than gay men or bisexual women (Table 1). Data from representative samples of the population, at least in the US, indicate the opposite should have been found. This limitation needs to be addressed, given its validity implications. In addition, protective factors are not covariates and, thus, should be excised from such a listing. Address here the issue of raw and final scores that are mentioned in Table 2 and indicate which is used in analyses.

Be careful in the Discussion section. You indicate that all SOGI groups differed from heterosexual peers. However, the means between lesbian/gay and heterosexual individuals are often rather similar. When differences arise, so, too, do validity concerns that circle back to the sample. For example, fewer gay than heterosexual men meet cut-off for depression (38.9% vs. 46.3%). You indicate this finding is not significant. Nevertheless, the effect size seems noteworthy and it is contrary to expectations.

Issues Concerning the Tables:

1. Table 2 is confusing. It can be improved by focusing on the raw or final scores, as indicated above. Also, add the corresponding t-tests, p values, and d statistics for the last 2 rows, as it seems reasonable to compare the SOGI groups to Brähler’s data. I imagine most individuals in Brähler were heterosexual; please clarify in the table note. By “overall” sample, I gather you mean the analytic sample.

2. A table addressing Hypothesis 2 is necessary. It would provide the ANOVA test for well-being, the chi-square test for the depression cut-off, their effect sizes, and identify the significant pairwise comparisons.

3. For the current Table 3, please add the regression statistics for the significant moderating effects concerning the protective factors. Test moderating relations of sex by SOGI on outcomes.

6. PLOS authors have the option to publish the peer review history of their article (what does this mean?). If published, this will include your full peer review and any attached files.

Reviewer #1: **Yes: **Carina Heckert

Reviewer #2: No

---

## [Author Response · Author response to Decision Letter 0]

3 May 2021

PONE-D-21-06928 Response to Reviewers

Editor Comments: The two Reviewers made two conflicting judgments. In my opinion, I believe that the breadth of the sample collected and the correctness of the experimental design merit giving the authors a chance to respond to the Reviewers and revise the manuscript accordingly. I therefore urge the authors to resolve all comments from the two Reviewers. In particular, I urge the authors to pay particular attention to the issues raised by Reviewer 2 related to methodology (recruitment information) and proper interpretation of the data (e.g., between lesbian/gay and heterosexual individuals). Finally, I recall the importance of what Reviewer 1 raised regarding the use of a standard English language, which is a prerequisite for publication. I suggest that authors either state that the language review was conducted by a native English-speaking author or that they use a professional proofreading service.

>>> Thank you very much for the opportunity to revise our manuscript. As suggested by the Editor, we have resolved all comments from the two Reviewers and in particular focused, as suggested by the Editor, on the revision of the Methodology (recruitment information) and the interpretation of the data. Further, we have asked a native English-speaking colleague to revise the language and added a statement in the acknowledgments. <<<

Journal Requirements: When submitting your revision, we need you to address these additional requirements.

>>> We have addressed the requirements and made changes accordingly. <<<

'Pichit Buspavanich received a research grant from Gilead. All other authors have no conflict of interest to declare. Sven Mahner reports Research support, advisory board, honoraria and travel expenses from AbbVie, AstraZeneca, Clovis, Eisai, GlaxoSmithKline, Medac, MSD, Novartis, Olympus, PharmaMar, Pfizer, Roche, Sensor Kinesis, Teva, and Tesaro.'

>>> Thank you very much for sharing your guide for authors. We confirm that our Competing Interest statement does not alter the adherence to PLOS ONE policies on sharing data and materials. Therefore, the Competing Interest statement will not need any change. With regard to data availability, we have agreed to share data without any restrictions. <<<

>>> As stated in the answer above, no change in the statement is necessary. <<<

3. We note that you have indicated that data from this study are available upon request. PLOS only allows data to be available upon request if there are legal or ethical restrictions on sharing data publicly. For information on unacceptable data access restrictions, please see http://journals.plos.org/plosone/s/data-availability#loc-unacceptable-data-access-restrictions. In your revised cover letter, please address the following prompts:

b) If there are no restrictions, please upload the minimal anonymized data set necessary to replicate your study findings as either Supporting Information files or to a stable, public repository and provide us with the relevant URLs, DOIs, or accession numbers. Please see http://www.bmj.com/content/340/bmj. c181.long for guidelines on how to de-identify and prepare clinical data for publication. For a list of acceptable repositories, please see http://journals.plos.org/plosone/s/data-availability#loc-recommended-repositories. We will update your Data Availability statement on your behalf to reflect the information you provide.

>>> Many thanks for pointing out these important restrictions. We have agreed to share data without any restrictions. Thank you for updating the statement of Data Availability on our behalf. We have uploaded our data set. <<<

>>> Thank you for pointing out that the ethics statement should only appear in the Methods section. We have revised our manuscript accordingly. <<<

Reviewer #1: The manuscript makes an incremental but important contribution to understanding mental health during the COVID pandemic. I have two suggestions for the authors. First, a short section on the research context would be useful to readers unfamiliar with Germany. For example, details on the German social safety net and how this may have offset some of the ill effects of the pandemic, such as unemployment, are things that may have a bearing on mental health status. If the German government had any responses to help mitigate the effects of the pandemic would also be relevant. Additionally, details on the legal and social climate toward gender and sexual minorities would be useful for understanding the mental health of this population. Such details about the local context could help show to what extent the results are generalizable in other locations. Second, the manuscript would benefit from a couple of rounds of careful editing. I noticed a number of grammatical errors and awkwardly worded passages that could use some cleaning up.

>>> Thank you very much for these important comments. We agree that it is of great importance to provide information on the country specific research context, in order to ensure that readers will understand the German context. As suggested, we have added a section on the COVID-19 related research context in Germany with regard to social safety and health (page 4 of the manuscript). Further, as suggested, we have added details on the legal and social climate toward gender and sexual minorities in Germany (page 6 of the manuscript). Further, we have edited the entire manuscript and had it proofread by our dear colleague Grace O'Malley, PhD in Clinical Psychology, who is a native English speaker. <<<

Reviewer #2: The manuscript would benefit from stronger grounding in minority stress theory. This would allow the authors, for example, to remove stress from health outcomes (p.4), to distinguish between protective factors and covariates, and to specify hypotheses pertaining to protective factors. The authors imply that SOGI individuals will be more stressed by the pandemic and, thus, experience even poorer mental health than before the pandemic. This is not testable, given their data. The authors sometimes conflate findings and issues pertaining to gender minority individuals with those of sexual minority individuals, a problem apparent in the Introduction and Discussion.

>>> Thank you very much for this suggestion. We agree that the manuscript would benefit from a stronger grounding in Minority Stress Theory. Therefore, we have added considerably more details on the theory in the Introduction section (page 5 and 6 of the manuscript). However, please be aware that our research question was not to examine the influence of Minority Stress on well-being rather than a comparison of the levels of well-being among all LGBTQIA* populations compared to a cis-heterosexual population, Thus, in regards to the protective factors we would like to kindly explain, that the selected protective factors are indeed protective factors for well-being based on a literature review for variables such as age (Busch, Maske, Ryl, Schlack, & Hapke, 2013; Erhart & von Stillfried, 2012), employment (Busch et al., 2013), partnership status (Bulloch, Williams, Lavorato, & Patten, 2017; Gariepy, Honkaniemi, & Quesnel-Vallee, 2016), place of living (Rommel, Bretschneider, Kroll, Prütz, & Thom, 2017; Thom, Kuhnert, Born, & Hapke, 2017), and children (Gariepy et al., 2016) during the COVID-19 pandemic. In our opinion, this selection is more suitable for our study since we do a comparison of cis-heterosexual and non-cis/non-heterosexual individuals and therefore we aim to find protective factors which are associated in all groups. The Minority Stress Theory focusses only on protective factors pertaining to the unique stress experienced by people with minoritized sexual and gender identities. To make that clear, we replaced the term “Socio-demographic characteristics and other covariates” to “protective factors for well-being” in the Methods and Statistical analysis sections of the manuscript. To acknowledge the valuable point of the Reviewer #2, we added a sentence in the Discussion section (page 22 of the manuscript). Further, we carefully scanned the manuscript for any conflation of sexual and gender minority individuals. The analyses are structured in a way that shows LGBTQI* people both as one group and the heterogeneity within the LGBTQI* group. The latter point especially targets the problem that sexual and gender issues are so often conflated in this literature. We have adjusted the wording where appropriate throughout the introduction and discussion section to reemphasize that this distinction is intended and meaningful. <<<

Reviewer #2: The authors imply that SOGI individuals will be more stressed by the pandemic and, thus, experience even poorer mental health than before the pandemic. This is not testable, given their data. The authors sometimes conflate findings and issues pertaining to gender minority individuals with those of sexual minority individuals, a problem apparent in the Introduction and Discussion.

>>> Thank you very much for this important comment. We agree with Reviewer #2 and have therefore already included a statement in the limitation section in the first version of our manuscript (page 24 of the manuscript). In our opinion with this limitation readers are well informed about the caution of that matter. Further, as already mentioned above, we carefully scanned the manuscript to adjust any wording that may conflate findings in regards of sexual and gender minority individuals. Thank you for drawing attention to this important point. <<<

Reviewer #2: The Method section would benefit from more recruitment information. As it reads, someone could not replicate the study. For examples, what chats were accessed on the social media sites and how was the study advertised to prospective participants. Such information may explain why the authors have more lesbians than gay men or bisexual women (Table 1). Data from representative samples of the population, at least in the US, indicate the opposite should have been found. This limitation needs to be addressed, given its validity implications. In addition, protective factors are not covariates and, thus, should be excised from such a listing. Address here the issue of raw and final scores that are mentioned in Table 2 and indicate which is used in analyses.

>>> Thank you very much for addressing a lack of recruitment information. We agree that the recruitment description could benefit from more information. We have therefore added considerably more details regarding the recruitment (page 8 of the manuscript).

Relatedly, we have also addressed the limitation with regard to the high participation rate of cis-women, which we observe among those identifying as heterosexual and lesbian. This skewed recruitment is likely a by-product of the advertisement text used. We have included this caveat in the Method section. Implications for validity were added to the Discussion section (page 23 of the manuscript). With regard to the comment on protective factors, we have already addressed this in our comment above. With regard to the comment on raw and final scores, we followed Reviewer #2’s suggestions below and we decided to delete the rare score from Table 2 (see comments below) and changed Table 2 as suggested by Reviewer #2. <<<

Reviewer #2: Be careful in the Discussion section. You indicate that all SOGI groups differed from heterosexual peers. However, the means between lesbian/gay and heterosexual individuals are often rather similar. When differences arise, so, too, do validity concerns that circle back to the sample. For example, fewer gay than heterosexual men meet cut-off for depression (38.9% vs. 46.3%). You indicate this finding is not significant. Nevertheless, the effect size seems noteworthy and it is contrary to expectations.

>>> Thank you very much for this valuable point. We completely agree and would like to point out that we already discussed this matter in a paragraph in our first version of the manuscript (page 21 of the manuscript). However, to make this point clearer, we have now added another sentence in the manuscript (page 20 of the manuscript). <<<

Reviewer #2 Issues Concerning the Tables: 

1. Table 2 is confusing. It can be improved by focusing on the raw or final scores, as indicated above. Also, add the corresponding t-tests, p values, and d statistics for the last 2 rows, as it seems reasonable to compare the SOGI groups to Brähler’s data. I imagine most individuals in Brähler were heterosexual; please clarify in the table note. By “overall” sample, I gather you mean the analytic sample.

>>> Thank you for these suggestions. As suggested, we have revised Table 2 (page 16 of the manuscript), deleted the finale scores and added t-tests, p-values and d statistics for the last two rows. Further, we added the table note. With “overall sample” we referred to the total sample of the present study, in order to differentiate between the overall sample and subgroups. However, to makes this clearer, we replaced “overall” with “total sample”. We believe Table 2 has benefited from these changes, many thanks for your suggestions. <<<

2. A table addressing Hypothesis 2 is necessary. It would provide the ANOVA test for well-being, the chi-square test for the depression cut-off, their effect sizes, and identify the significant pairwise comparisons.

>>> We thank Reviewer #2 for the suggestion to include another table with further analyses for testing Hypothesis 2 (Table 3, page 17 of the manuscript). However, since these are two different dependent variables (i.e., well-being and the dichotomous variable above/below cut-off score) we have found a compromise: The recommended ANOVA for well-being as well as posthoc tests are already reported in the text (see page 14). The mean values can be found in Table 2. In addition, as recommended by Reviewer #2, we now have conducted a chi-square test for the depression cut-off. The chi-square test showed a significant p-value (p < 0.001). To analyze which groups had a higher likelihood of being below the cut-off score, we performed logistic regression and included the results in a new table (Table 4, page 18 of the manuscript). <<<

3. For the current Table 3, please add the regression statistics for the significant moderating effects concerning the protective factors. Test moderating relations of sex by SOGI on outcomes.

>>> Thank you very much for this interesting point. As requested, we added Table 3 to include interaction results of the protective factors (page 17 of the manuscript). In addition, we have now included the moderation effect in the abstract. Further, we thank Reviewer #2 for the idea of testing sex as a moderator. In our opinion, in this cohort of individuals of minoritized gender identities, the variable sex must be analyzed with caution. In our study the variable sex was obtained in a self-assigned assessment. We used this information to assign into the different gender and sexual identities: (I) cis-heterosexual women, (II) cis-heterosexual men, (III) cis-lesbian women, (IV) cis-gay men, (V) cis-bisexual women, (VI) cis-bisexual men, (VII) cis-asexual women, (VIII) cis-asexual men, (IX) trans* women, (X) trans* men, (XI) non-binary gender identities (participants who identify as female and male), and (XII) inter* people. This information can be found under Measures in the Methods section. In our opinion, we would not gain more information with the moderation analysis of the identity and the sex. However, we performed this moderation analysis and revealed no significant effect for sex as moderator (p = 0.912), as expected. Therefore, we decided not to include this information in the revised manuscript as we believe it to be redundant. However, if the editor believes it is important to include this information, we are willing to present it. Please be aware that in this case the correct term would be “self-assigned gender”, which would also prevent any discrimination of the vulnerable group of minoritized gender identities. <<<

---

## [Decision Letter · Decision Letter 1]

14 May 2021

Well-being during COVID-19 pandemic: A comparison of individuals with minoritized sexual and gender identities and cis-heterosexual individuals

PONE-D-21-06928R1

Dear Dr. Buspavanich,

We’re pleased to inform you that your manuscript has been judged scientifically suitable for publication and will be formally accepted for publication once it meets all outstanding technical requirements.

Kind regards,

Stefano Federici, Ph.D.

Academic Editor

PLOS ONE

Additional Editor Comments (optional):

Reviewers' comments:

Reviewer's Responses to Questions

**Comments to the Author**

1. If the authors have adequately addressed your comments raised in a previous round of review and you feel that this manuscript is now acceptable for publication, you may indicate that here to bypass the “Comments to the Author” section, enter your conflict of interest statement in the “Confidential to Editor” section, and submit your "Accept" recommendation.

Reviewer #1: All comments have been addressed

2. Is the manuscript technically sound, and do the data support the conclusions?

Reviewer #1: Yes

3. Has the statistical analysis been performed appropriately and rigorously? 

Reviewer #1: Yes

4. Have the authors made all data underlying the findings in their manuscript fully available?

Reviewer #1: Yes

5. Is the manuscript presented in an intelligible fashion and written in standard English?

Reviewer #1: Yes

6. Review Comments to the Author

Reviewer #1: The authors have addressed my previous concerns, which has led to significant improvement and clarity in the manuscript.

7. PLOS authors have the option to publish the peer review history of their article (what does this mean?). If published, this will include your full peer review and any attached files.

Reviewer #1: **Yes: **Carina Heckert

---

## [Editor Report · Acceptance letter]

21 May 2021

PONE-D-21-06928R1 

Well-being during COVID-19 pandemic: A comparison of individuals with minoritized sexual and gender identities and cis-heterosexual individuals 

Dear Dr. Buspavanich:

I'm pleased to inform you that your manuscript has been deemed suitable for publication in PLOS ONE. Congratulations! Your manuscript is now with our production department. 

Kind regards, 

on behalf of

Prof. Stefano Federici 

Academic Editor

PLOS ONE